# Finding women in fishing communities around Lake Victoria: "Feasibility and acceptability of using phones and tracking devices"

**Gertrude Nanyonjo**[1]*, **Zachary Kwena**[2], **Sarah Nakamanya**[3], **Elialilia Okello**[4], **Bertha Oketch**[2], **Ubaldo M. Bahemuka**[3], **Ali Ssetaala**[1], **Brenda Okech**[1], **Matt A. Price**[5,6], **Saidi Kapiga**[4,7], **Pat Fast**[5], **Elizabeth Bukusi**[2], **Janet Seeley**[3,7], **the LVCHR study team**[¶]

1 UVRI-IAVI HIV Vaccine Program Limited, Entebbe, Uganda, 2 Research Care and Training Program (RCTP), Kenya Medical Research Institute, Kisumu, Kenya, 3 Medical Research Council/Uganda Virus Research Institute and London School of Hygiene and Tropical Medicine (MRC/UVRI & LSHTM) Uganda Research Unit, Entebbe, Uganda, 4 National Institute for Medical Research, Mwanza Intervention Trials Unit (MITU), Mwanza, Tanzania, 5 IAVI, New York, NY, United States of America, 6 Department of Epidemiology and Biostatistics, University of California at San Francisco, San Francisco, CA, United States of America, 7 London School of Hygiene and Tropical Medicine, London, United Kingdom

¶ Membership of Lake Victoria Consortium for Health Research (LVCHR) Team is provided in the acknowledgment.

* gnanyonjo@iavi.or.ug

**Data Availability Statement:** All relevant data are within the paper.

# Abstract

## Introduction

Women in fishing communities have both high HIV prevalence and incidence, hence they are a priority population for HIV prevention and treatment interventions. However, their mobility is likely to compromise the effectiveness of interventions. We assessed the acceptability, feasibility and of using phones and global positioning system (GPS) devices for tracking mobility, to inform future health research innovations.

## Methods

A mult-site formative qualitative study was conducted in six purposively selected Fishing Communities on the shores of Lake Victoria in Kenya, Tanzania, and Uganda. Participants were selected based on duration of stay in the community and frequency of movement. Sixty-four (64) women participated in the study (16 per fishing community). Twenty-four (24) participants were given a study phone; 24 were asked to use their own phones and 16 were provided with a portable GPS device to understand what is most preferred. Women were interviewed about their experiences and recommendations on carrying GPS devices or phones. Twenty four (24) Focus Group Discussions with 8–12 participants were conducted with community members to generate data on community perceptions regarding GPS devices and phones acceptability among women. Data were analyzed thematically and compared across sites/countries.

**Funding:** This work was funded in part by IAVI and made possible by the support of many donors, including United States Agency for International Development (USAID). A list of IAVI donors is available at http://www.iavi.org.

**Competing interests:** The authors have no competing interests.

## Results

Women reported being willing to use tracking devices (both phones and GPS) because they are easy to carry. Their own phone was preferred compared to a study phone and GPS device because they were not required to carry an additional device, worry about losing it or be questioned about the extra device by their sexual partner. Women who carried GPS devices suggested more sensitization in communities to avoid domestic conflicts and public concern. Women suggested changing the GPS colour from white to a darker colour and, design to look like a commonly used object such as a telephone Subscriber Identity Module (SIM) card, a rosary/necklace or a ring for easy and safe storage.

## Conclusion

Women in the study communities were willing to have their movements tracked, embraced the use of phones and GPS devices for mobility tracking. Devices need to be redesigned to be more discrete, but they could be valuable tools to understanding movement patterns and inform design of interventions for these mobile populations.

## Introduction

HIV remains a major problem in sub-Saharan Africa where the rate of new HIV infections is still unacceptably high despite the increased availability of proven interventions. By the end of 2021, an estimated 37.7 million people globally were living with HIV, while 1.5 became newly infected in the same year despite the progress in treatment [1]. Studies conducted in fishing communities of Lake Victoria in East Africa have reported high HIV prevalence and incidence, with more than half of the women infected with HIV in some age groups [1]. Moreover, in these settings access to health care services is limited [2–5].

Many women in fishing communities are very mobile; they change places of work and sexual partners frequently, which results in increased risk of acquisition of HIV and other sexually transmitted infections [6,7]. Mobility can also interfere with research that requires follow up visits. Clinical trials require high levels of adherence to scheduled study visits especially with the use of investigational products. Participant retention (follow-up rates) affect the validity of study outcomes, the statistical power to detect meaningful differences, and the evaluation of the efficacy and effectiveness of interventions [8,9]. High mobility of participants that takes enrolled participants far away from the study catchment area is one of the obstacles to retention in clinical trials [10].

Population mobility affects engagement in both HIV care and prevention as a barrier to both individual access to health care facilities and the ability to determine if an individual is truly lost from care. Individuals who drop out of care at one facility may continue at a second facility [11] with or without the knowledge of the facility where they were initially enrolled. Similarly, studies on retention in HIV care show that tracking an individual in care can facilitate their continued care, and this could address the difficulties of trying to maintain both contact and care while traveling [12]. Similar to clinical care, public health intervention may require regular follow up of clients to complete their scheduled visits and/or treatment [13,14]. Mobility has been associated with high risk behaviour and poor access to healthcare services [15,16]. Mobility can disrupt individuals' access to HIV care systems, disconnecting them from care and treatment, and leading to poor health outcomes [17,18].

Women's mobility in fishing communities is mainly driven by the search for economic opportunities, although others travel to join their families, seek medical care, and for leisure or to get married [19]. While fishermen follow fish movements to secure a good fish catch, women fish traders, and even those involved in fish processing activities, such as scaling, drying, and smoking, in turn follow the fishermen [15,20]. Thus the women's mobility patterns are thought to revolve around fish-landing beaches, retail markets and their rural homes [7,15,19,20]. To understand women's mobility requires collecting empirical and accurate data on their characteristics, their routes and timing of travel [21].

Tracking women's movements to understand the nature, patterns and routes followed is important to support the designing of studies that put in place mechanisms to minimize loss to follow up and enhance retention rates. While tracking women may be valuable for research, it is not yet clear how women in fishing communities would perceive the use of tracking devices and what their experience with the devices may be. Thus, in this study we assessed the perceptions and thoughts of women about the use of phones and global positioning system (GPS) devices used as tracking devices to monitor the extent and patterns of mobility in the fishing communities of Lake Victoria in Kenya, Uganda and Tanzania. This study was implemented by Uganda Virus Research Institute-International AIDS Vaccine Initiative HIV Vaccine Program (UVRI-IAVI), Kenya Medical Research Institute (KEMRI), Medical Research Council/Uganda Virus Research Institute & London School of Hygiene and Tropical Medicine, Uganda Research Unit (MRC/UVRI and LSHTM) and, the Mwanza Intervention Trials Unit and Tanzania's National Institute for Medical Research (MITU/NIMR) under the umbrella the Lake Victoria Consortium for Health Research (LVCHR).

## Methods

### Study design

This was a multi-site formative qualitative study conducted between February and June 2018 among members of fishing communities of Lake Victoria in Kenya, Uganda and Tanzania. This study used in-depth interviews (IDIs) to document the perceptions, and experiences of women in fishing communities who carried either a GPS device or a phone. Besides, community FGDs were considered appropriate for generating data on community perceptions regarding GPS devices and phones acceptability among women. The discussions were held with residents of FCs (young women and men 18–25 years of age, older women, and older men 26 years+) separately to explore their views on the acceptability of using GPS devices and phones by women. During the discussions, participants were also asked to suggest how best individuals could be identified and approached to participate in carrying the devices. IDIs were considered appropriate for generating data on personal lived experiences of those who actually carried the GPS devices, their own phones and study phones handed out by the researchers. The GPS devices were white in colour, smaller than a fist in size and could easily be kept in a hand bag. The phones were basic inexpensive mobile phones loaded with weekly credit of roughly 1.5 US dollars. The long-term goal was to develop methods which could be used to measure the extent and patterns of mobility among women residents and/or working in fishing communities. However, the immediate goal was to assess the acceptability and feasibility of using tracking devices (phones and GPS gadgets) that could support tracking the participant's movements to inform the design of large intervention studies.

### Study settings

The study was conducted in six purposively selected fishing communities with a known HIV prevalence of > 15% among women and population of at least 1000 people, on the shores of

Lake Victoria in the three East African countries of Kenya, Uganda and Tanzania. Two communities were selected from each country. This was a multi-site qualitative study conducted by investigators from the partners of the Lake Victoria Consortium for Health Research (LVCHR). LVCHR members include: i) Kenya Medical Research Institute (KEMRI); ii) Mwanza Intervention Trials Unit (MITU) in Tanzania; iii) International AIDS Vaccine Initiative unit within the Uganda Virus Institute (UVRI/IAVI); and iv) the Medical Research Council/Uganda Virus Research Institute Uganda Research Unit & London School of Hygiene and Tropical Medicine (MRC/UVRI & LSHTM) Uganda Research Unit [22]. Within the context of this study, a fishing community consisted of one or more landing sites on either mainland or island where people were focused on fishing and fishing-related activities and lived together in a defined geographical area. In some cases, these communities coincided with administrative boundaries such as villages and wards. Landing sites in these FC are often very busy, attracting a large number of people who come to engage in fish-related activities. As a result, there is often intense competition for traders to access fish for markets. Overall, fishing community residents are known to exhibit high HIV risk behaviour. Although our study population comprised of women residents and /or working in fishing communities, men were also involved during entry meetings and Focus group community discussions where shared their views about the feasibility of using GPS and phone devices, identifying and how to approach women to participate in the study.

## Sampling procedure

The process of identifying participants started by conducting community entry meetings where we met the gatekeepers (both male and female) to obtain their support and buy-in. During community entry meetings, local leaders were requested to help in identifying women aged 18–45 years willing to carry GPS or phones and to help identify private and confidential meeting venues as well as mobilize potential participants. The study team screened the identified potential participants for uncoerced willingness to participate in the study as well as eligibility based on age above 18, at least 6 months residence in the community, own a phone and frequency of mobility. The eligible women were then scheduled for consenting and issuing of the study devices. On the scheduled date, the women were given detailed information about the study that aims to describe the feasibility of using mobile phones and GPS devices among women resident and/or working at the fishing communities and assess the feasibility of following them in a future health research innovations in Kenya, Tanzania and Uganda. Women were informed that they were selected to participate in this study because they lived in this community and had very important opinions regarding mobility patterns and the best ways of tracking and retaining women to participate in future health research innovations. Women were consented, enrolled into the study and randomized to one of the three arms: study phone, GPS device (Figs 1 and 2) or using own phone.

## Data collection and management

The Consortia leadership conceived and designed the study. Each site had one senior social science/behavioral researcher who coordinated activities of the study. These comprised of protocol training, supervision of field teams in data collection and management activities.

There were four field researchers at each site, both male and female field researchers of various ages (ranging from 23 to 45), who were selected from the research teams at each of the participating institutions based on their experience and training (see Fig 3). The field researchers had enough experience to conduct research in the study settings. The research teams with over five years' experience in collecting qualitative data were trained on the approach to data

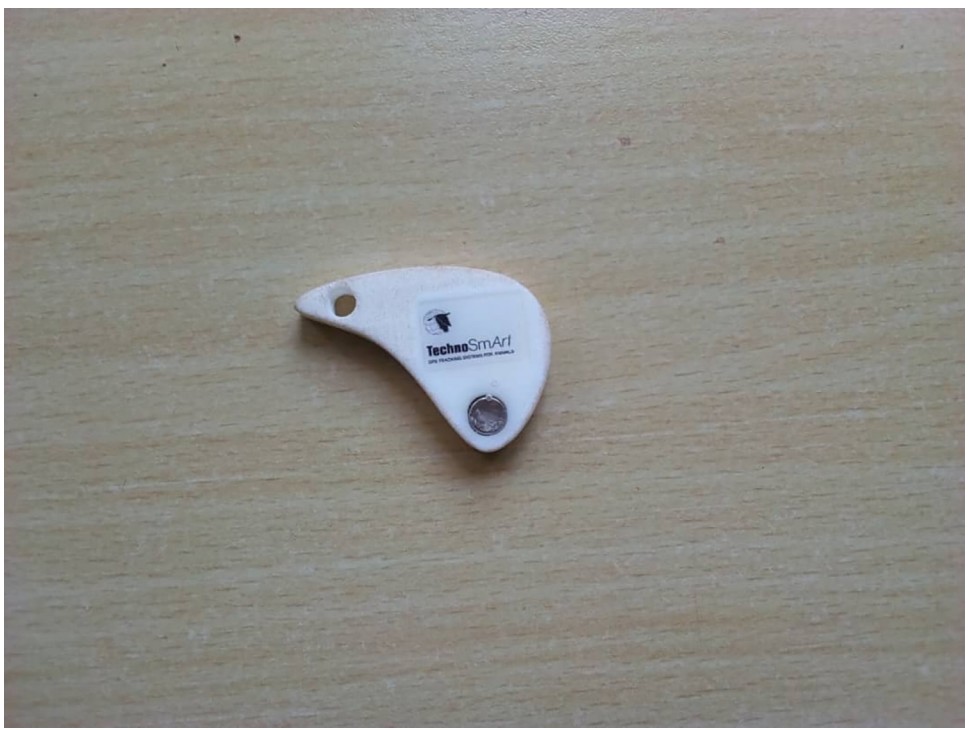

**Fig 1. Global positioning system (GPS) devices switch.**

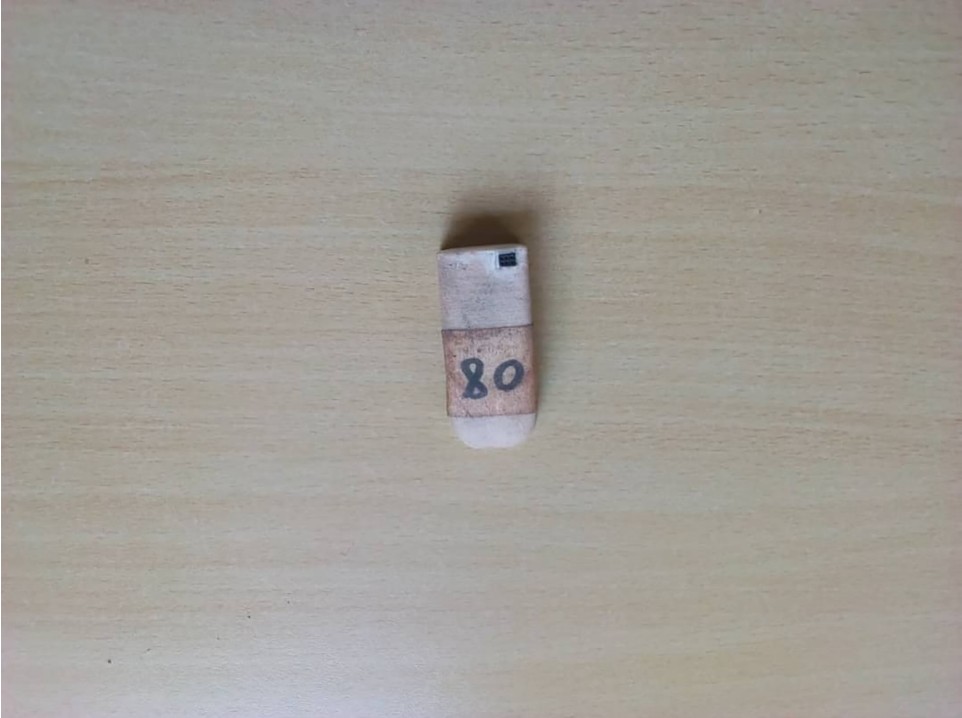

**Fig 2. Global positioning system (GPS) devices.**

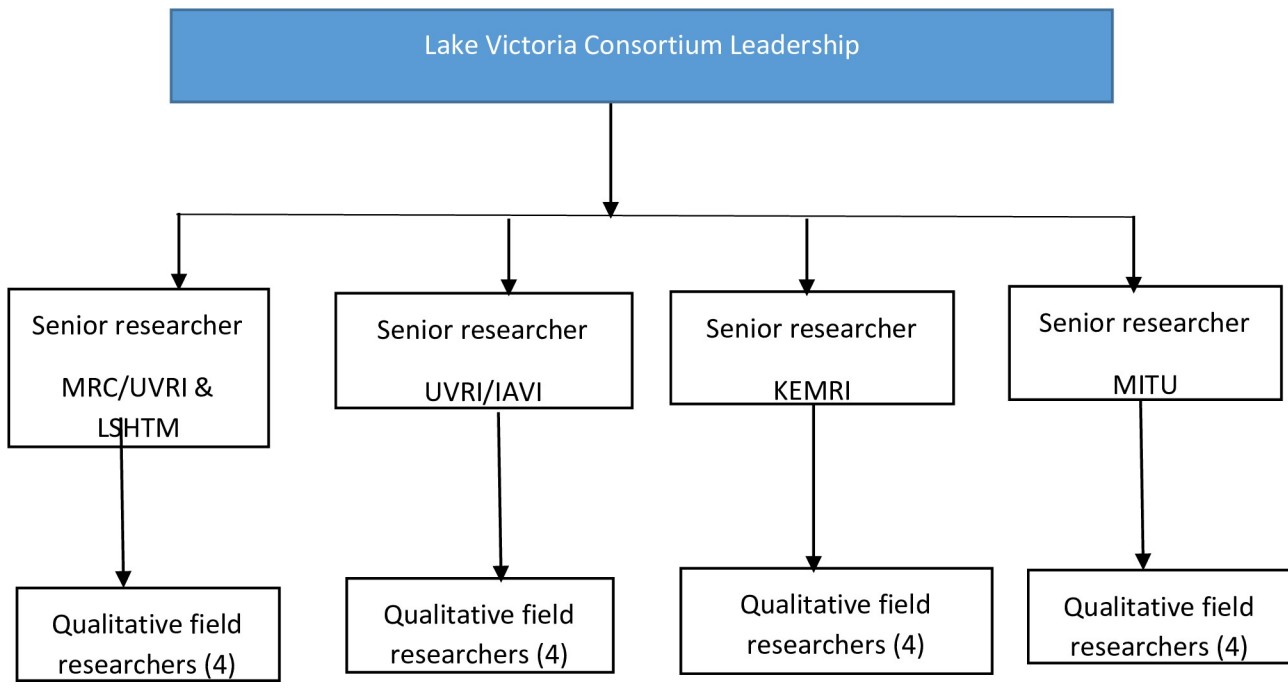

**Fig 3. Flowchart on structure of the research teams.**

collection including how to moderate Focus Group Discussion, Conduct In-depth interviews, the use of mobile phones, and GPS devices, and ethical considerations. After the training, research teams pre-tested the approaches and practiced on the use of GPS in a near-by fishing community that was not to be included in the main research. We organized and conducted to 6 introductory meetings with community gate keepers and the community members in each of the six fishing communities. Each meeting consisted of 8–15 members who had lived in the fishing communities for at least 1 year. Those who participated in the introductory sessions were nominated by local leaders in the communities. During community meeting, contact persons were identifies to support in mobilizing individuals to part in the FGDs and those to carry GPS device and phones. We conducted 4 Focus Group Discussions in each of the fishing community to explore views from different groups of people on the acceptability of using GPS devices and phones by women, how communities would think about GPS device and how to identify the women who are mobile to participate in carrying GPS device. Group discussion participants were identified in two ways: (1) selected by community local leaders and, (2) referrals from introductory meetings of people who were knowledgeable on the subject of discussion. The groups were divided into four categories based on gender and age: young women (18–25), older women (25+), young men (18–25) and older men (25+). Basing on the suggestions from the community meetings, and FGDs, the contact persons identified during community meetings supported the process of mobilizing women to carry GPS and phones. The identified women were invited to participate in the study. One group of 24 women, 6 from each implementing research institute were randomized to simple study mobile phone provided by the research institute that included credit for telephone use and another similar group of 24 were asked to use their own mobile phones or a mobile phone they have access to and given roughly USD 1.5 worth of credit for telephone use. Each woman from the groups that received/or used their own phones was contacted by phone after three, six and nine days by the research team and asked about any movements made outside the community they live

**Table 1. Number of participants assigned to each category of gadgets.**

| Gadgets | Category | | Total |
|---|---|---|---|
| GPS devices Arm | Women who carried GPS while in the community | 8 | |
| | Women who carried GPS while moving out of community | 8 | 16 |
| Phone Arm | Carried Institutional Phones | 24 | |
| | Carried own phones | 24 | 48 |
| | Total | | 64 |

in over the past 3 days. In addition, face-to-face IDIs were conducted monthly. In addition, a total of 16 women, 4 from each implementing site in Kenya, Tanzania and Uganda were given a small portable GPS device to carry for 4 months (see Table 1) line 210. Women were instructed on how to hand the device and how often the research would collect the devices to download data and charge the GPS. Based on the choice made during community discussions, four women from each site were identified to be invited to take part in the study.

Women were instructed on how to handle the device and how often the research would collect the devices on a monthly to download data and charge the GPS device. Women were met in a specified place on a monthly basis for 4 months and asked about their wellbeing, movements within the preceding 30 days and about their experience with carrying the device. Specifically, we collected data on: challenges and facilitators of carrying GPS/Phone, reactions from other people, battery issues as well as movement in and out of the communities. We also asked the participants how they wanted the GPS device modified to enhance convenience of carrying them. All interviews were done in a private place convenient to both the participant and interviewer. We conducted a total of 16 individual interviews with women who carried GPS: four in each research site in Kenya, Tanzania and Uganda, each interview lasting an average approximately 30 minutes. All interviews were conducted in the appropriate country/region local languages in either Kiswahili, Dholuo and Luganda, and later translated/transcribed into English. Detailed notes were captured and expanded to scripts after the discussions to avoid the fears associated with using an audio recorder [23].

## Data management and analysis

Notes taken during individual interactions/interviews were written out into expanded scripts and then saved in password-protected files on study computers. An effort was made to maintain participants' voices as quotes on interesting information that came up during the interview. Additionally, the files were backed up on encrypted external hard drives and kept off-site for security reasons. We started analysis by scanning through detailed scripts from interaction/interview notes using a data analysis template developed and agreed upon by the research team. The template guided development of codes and identification of emergent themes. The data sets from the different sites were merged to produce a single worksheet, which allowed easy navigation through, and comparison of the data from the different sites. Generation of themes was based on the topics covered in the interview guides. We also incorporated data-led analysis, allowing analytical themes to emerge from the scripts during the process of reading, exploration and coding responses. Four teams manually coded the scripts and shared the coding reports. We used the constant comparison technique to ensure consistency in coding across the sites.

## Inclusivity in global research

Additional information regarding the ethical, cultural, and scientific considerations specific to inclusivity in global research is included in the Supporting Information (SI Checklist).

### Ethical approval of the study and consenting

The study protocol was approved by Ethics Review Committees in the three countries: Uganda Virus Research Institute Research Ethics Committee (UVRI REC, #605) & Uganda National Council for Science and Technology (UNCST) (SS #4470) in Uganda, and KEMRI Scientific and Ethics Review Unit in Kenya (SERU, #3593) and the National Health Research Ethical Committee (NatHREC) (NIMR #2654) in Tanzania. All identified participants were given details about the study and all provided written informed consent before they were recruited in the study. We did not audio-record the discussions but instead took detailed notes and expanded them soon after the interview/discussion. This is because we wanted, to the extent possible, to collect data from participants while going about their usual business such as working or resting after their day's activities. Also, there have been concerns about audio-recording in some settings that diminish the quality of data collected [23]. This approach allowed us to ensure that the participants had discussions in a natural setting devoid of fears often associated with a audio recorder.

## Results

### Study participants characteristics

We enrolled 64 women (16 per site) between 18–45 years of age in fishing communities of Kenya, Uganda and Tanzania and conducted community FGD with different groups of people to seek their views on the use of the two devices. Majority of the women who participated in the study had attained primary level education, their main occupation was sex work and others engaged in fishing related activities including; fishing, fish mongering, fish processing and/or fish smoking and fifty percent of the participants reported being married (Table 2).

**Table 2. Participant characteristics.**

| Variables | MRC/UVRI and LSHTM | | UVRI-IAVI | | MITU | | KEMRI | | Total |
|---|---|---|---|---|---|---|---|---|---|
| | Phone users | GPS users | Phone users | GPS users | Phone users | GPS Users | Phone users | GPS users | N = 64 |
| **Sex** | | | | | | | | | |
| Male | 0 | 0 | | 0 | | 0 | | 0 | 0 |
| Female | 12 | 4 | 12 | 4 | 12 | 4 | 12 | 4 | 64 |
| **Age (years)** | | | | | | | | | |
| 18–24 | 5 | 2 | 3 | 1 | 2 | 2 | 2 | 1 | 18 |
| 25–34 | 3 | 1 | 5 | 2 | 6 | 1 | 5 | 2 | 25 |
| 35+ | 4 | 1 | 4 | 1 | 4 | 1 | 5 | 1 | 21 |
| **Education** | | | | | | | | | |
| None | 1 | 0 | 0 | 0 | 0 | 0 | * | * | 1 |
| Primary | 5 | 3 | 8 | 2 | 6 | 2 | * | * | 26 |
| Secondary | 4 | 1 | 2 | 2 | 4 | 2 | * | * | 15 |
| Tertiary | 2 | 0 | 2 | 0 | 2 | 0 | * | * | 6 |
| **Occupation** | | | | | | | | | |
| Fish related activities | 4 | 1 | 3 | 2 | 5 | 1 | 2 | 2 | 20 |
| Trader | 1 | 1 | 2 | 1 | 2 | | 5 | 1 | 13 |
| Sex/bar worker | 4 | 2 | 5 | 1 | 3 | 2 | 2 | 1 | 20 |
| Other | 3 | 0 | 2 | 0 | 2 | 1 | 3 | 0 | 11 |
| **Marital status** | | | | | | | | | |
| Married/cohabiting | 8 | 2 | 6 | 2 | 5 | 1 | 5 | 2 | 31 |
| Separated/single | 4 | 2 | 6 | 2 | 7 | 3 | 6 | 2 | 32 |
| Widowed | 0 | 0 | 0 | 0 | 0 | 0 | 1 | 0 | 1 |
| * Data not captured: FGD-Not collected | | | | | | | | | |

**Table 3. Experience and perceptions of GPS and phone users.**

| Themes | GPS user's issues | Phone user's issues | |
|---|---|---|---|
| | | Study phone users | Personal (own) phone users |
| 1.Willingness to participate in the study | 1.1 Sensitize communities on the benefits of GPS to the community and research institutions<br>1.2 Involve primary sexual partners before introducing GPS. | 1.1 Phone are easy to use<br>1.2 Preference to individual phones<br>1.3 Involve primary sexual partners before giving out phones or requesting women to use their own phones.<br>1.4 Preference of phones with two lines | 1.1 Involve primary sexual partners before giving out phones or requesting women to use their own phones.<br>1.2 Phone are easy to use |
| 2. Design of the GPS | 2.1 Design is user friendly<br>2.2 Created curiosity among community members<br>2.3 Make the design like a SIM card, rosary or ring like and should be waterproof. | | |
| 3.Security | 3.1 Gadgets being stolen due to mobility among women<br>3.2 Fear for the gadget to drown in water.<br>3.3 Confusing GPS with police work/investigations<br>3.4 Fear to lose the gadget<br>3.5 Fear of partners taking it way or hiding due suspicions<br>3.6 Fear of the gadget to be destroyed or damaged by children | 3.1 Fear that study phones may be stolen<br>3.2 Phones always fall in water<br>3.3 Sexual partners owning the phones | 3.1 Sexual partner picking the call and discover one is participating in research leading to breach of confidentiality |
| 4.Fearsworries/ uncertainties | 4.1Worries of inconsistent power supply<br>4.2 Worry of poor adherence to instruction<br>4.3 Worry of being spied by researchers or researchers using GPS to capture/ spy individual activities.<br>4.4 Gadget for sex workers who move a lot<br>4.5 Worry of stigma attached to GPS<br>4.6 Worry of spouses refusing to allow their partners to carry GPS<br>4.7 Fear of family conflict<br>4.8 Worry of explaining how the gadget works to spouses or family members<br>4.9 GPS could be left home intentionally due to fear of theft | 4.1 Worry of inconsistent power supply<br>4.2 Worry of network challenges<br>4.3 Worry of violence in households that could result out of having more than one phone.<br>4.4 Burden of carrying two phones<br>4.5 Fear to operate a SMART phone for those who are illiterate. | 4.1 Worry of network challenges<br>4.2 Worry of inconsistent power supply |

Main findings have been grouped into five thematic areas including: (a) willingness to carry GPS/phone, (b) design of GPS (c) security, (d) fears, worries and uncertainties around the device, (e) experiences of those who carried GPs and phones. We provide an overview of the findings in Table 3.

## Willingness to participate in the study

Majority of the women who were approached indicated their willingness to participate in the study. Participants expressed interest because they were highly mobile in search of business, health services and following fish based on seasons. The women also perceived positively the idea of using GPS device and phone to track their movement. Participants, however, emphasized the need to sensitize communities and thereby inform their spouses/sexual partners about the devices and therefore not raise any suspicions

> "*I will be willing to participate and I am very sure even other women if educated/sensitized and understand how the GPS device works, I am pretty sure they will be willing to take part in the study. Provide detailed information about the study to the community and women to clearly understand the benefits*" IDI Tanzania

Readiness to use their own and study phones was high. Many of the participants preferred their own phones and those who used study phones suggested the need to involve their sexual partners in providing information on the study and in the decision for them to take part to prevent concerns that could arise from having a new phone or two phones. Some participants were of the view that providing phones with dual lines that can accommodate both their personal line and the study's line was preferred. The issue of network connectivity was pointed out that some communities have limited access to cell phone networks. Some mobile service providers had stronger networks in certain areas while others had weaker networks depending on the location of their communication mast.

"*Use of phones to trace mobile women is the best because, if you have the contact of the person, you can easily reach out to her at any time, make reminders*" IDI Uganda

Although women were willing to participate in the study, there was need to demystify the project and counteract myths and misconceptions that the community members and women in particular would have about their participation in the study. Those who participated in the Focus Group Discussion emphasized the need to sensitize the communities, women and their primary partners about the benefits of using the devices (phone and GPS) and how communities and individual participants would benefit for instance accessing health interventions. It was pointed out that people would be worried that the research team would spy and know the activities they were involved in. Participants suggested the need for detailed information about the exact information the gadget would collect.

"*Encourage women participating in the study to open up to their primary partners, sit with the husbands/partners and explain to them why you are doing the tracking to help them understand the aim of using GPS as a good technology*" (Young Men FGD, Kenya).

## Design of global positioning system

Women were happy that the design of the GPS was user friendly but were worried that they would lose the gadget or even a partner/spouse would be suspicious and take it away. However, majority of the women suggested the need to reduce the GPS size to a size of a Subscriber Identity module (SIM) card or finger ring.

"*The GPS is small, I had no big challenge carrying it because I could even put it in my bra. However, those with difficulty carrying GPS should be given phones*" (IDI, Uganda).

However, some suggested that if the size of the GPS is reduced to a size like a SIM card or a ring, and made water proof, it would not be seen by their partners. The worry about inadvertently soaking them in water would be eliminated.

"*Make it like a SIM card or ring and water proof because majority of the women move on water. If by mistake it falls in water, it may not be damaged*" (IDI, Kenya).

## Security

Women were concerned about safety of the GPS devices, how to keep them securely since they travel a lot on water and also children could spoil them.

"*The concern in carrying the device is that it would get wet if not covered, could get lost, drown and could be destroyed by children*" (IDI, Tanzania)

On the other hand, the phone users reported that using both study and own phones was a good idea. However, they did have some reservations due to fear of the challenges that could come with the handling of phones, especially study phones such as theft of the phone as women went about their usual business.

"*During such travels one's phone can be stolen and the houses we stay in are not so safe. If one hears that you received a smart phone for instance, chances of stealing the phone from you are high. It's better to work with individual [own] phones*" (IDI, Uganda).

### Fears, worries and uncertainties around the devices

When participants were asked about the challenges of carrying a GPS device. Some felt unsafe with GPS, especially when they are not sure of what it is recording. Some participants felt that adhering to instructions of carrying the GPS all the time they move is very hard. Other felt that their sexual partners would question why they are carrying such devices. Some were scared of losing the GPS. On the other hand, the phone users in both arms presented issues of inconsistent power supply, network challenges. Those who were provided with institutional phones expressed the burden of carrying two phones and losing the institutional phone.

"*The biggest worry would be my partner failing to understand the importance of the GPS gadget. Sometimes I leave home for business but get involved in other activities like checking on friend/sexual partners. If I carry that gadget it may get lost*" (IDI, Uganda).

### Concern about male involvement

Women stressed the need to involve their sexual partners at time of issuing phones to relieve them the burden of explaining/anawering questions, sensitize them on to the use of devices and also involving them in future studies since they are mobile too. Participants suggested that men should also be tracked since they are mobile and are potential participants for clinical trials.

"*Involving male sexual partners is crucial, initially, I would forget to carry GPS but I had informed my sexual partner about the study which he was comfortable with. Whenever I would leave the gadget behind, he would remind me*" (IDI, Kenya).

Participants recalled that phones have led to family instability in the past. For instance, if a woman receives a call in the presence of the partner, he would want to know who has made the call and the reason for the call. If it is the study providing the phones, participants should be encouraged to let their male partners know about their study participation and requirements.

"*Some women are not trusted by their sexual partners, he may destroy the phone if not informed prior as to why his partner is holding a second phone. Others may grab the phone away for from their partners basing on circumstances prevailing at that time for instance, the male partner can give it out to another sexual partner as observed in our settings*!" (FGD Young women, Uganda).

*To prevent conflict that might come with project phones, there should be need to engage male counterparts to help them understand the reason why his partner is being give[n] a phone. If the male sexual partner is suspicious then the woman should be left to carry her own phone and be provided with airtime* (IDI, Tanzania).

Women who carried GPS felt discomfort due to the white colour because they would get dirty easily. The women were particularly concerned about community members wondering about the purpose of the small gadgets the women were carrying. The women suggested that one needs a safe and comfortable place to carry the GPS devices. Others had a fear of being seen by male partners who they thought may wrongly associate the device with financial gain.

"*It was very difficult for me to [in] explain to my sexual partners why I was carrying the GPS device and how it operates. Married women like me could have similar challenges of explaining to their husbands about the gadget which may lead to suspicion that may be I am connecting to other men*" (IDI, Uganda).

Other women described the device as very small, not heavy to carry and felt that they were able to move with it everywhere they travelled. Women said the device was user friendly. They requested for small bags where they can keep them while moving with them.

"*The GPS is easy to carry, it only got dirty because I did not have a proper way of carrying it*" *(IDI, Uganda).*

For the phone users, it was noted that it could be a burden carrying two phones.
Other women reported that they were labelled in the community as spies and were believed to have been given or bribed with a lot of money to carry the gadgets. Some women shared that they found it challenging to explain to their clients what the devices were and the reason for carrying them.

"*Use of phones to trace mobile women is the best, however, some islands have no electricity, others have network challenges, there is need to understand which communities/landing site/ islands have better networks such that women can have either both MTN and Airtel lines to curb the challenge of network. In such communities, the GPS would solve the problem too*" *FGD Uganda*

The fear of violence in households manifested itself in the form of men's complaints that they were "not informed" and did not have time to attend meetings that were organized at the beginning of the study. In such cases, men tend to ask their sexual partners where they get the money to buy more than one phone, thinking that they have other sexual partners. In one particular case, one woman described her sexual partner as a "selfish man":

"*Again men think that they are the only ones to own nice items like phones, if a woman works hard, you hear them blaming women to have multiple sexual partners. Men want to snatch women's phones such that other men do not reach out to them*" (IDI, Uganda).

Women noted that GPS device and phones are very key tracking methods that if communities are sensitized on the importance of the devices, women would have no issues to move with them.

## Discussion

In this study we assessed the acceptability and feasibility of using GPS devices and phones to measure mobility among women residents/working at fishing communities along Lake Victoria in Uganda, Kenya and Tanzania. We found that participants were willing and accepted the use of both phones and GPS devices. However, they were much more willing to use phones, especially personal phones, compared to using a GPS device since they are used to their own phones. We also found that if a GPS device was re-designed to make them much smaller and take the shape of everyday use items like rings and/or SIM cards this would make them much more appealing.

Overall, we found that it was feasible and acceptable to use GPS devices among our sample of women. Participants enjoyed carrying the GPS devices. Our findings are similar to those from a study that evaluated the feasibility and acceptability of using GPS methods to understand the spatial context of substance use and sexual risk behaviours among a sample of Young Men who have Sex with Men in New York City [24]. Participants in this study reported high ratings of pre-GPS acceptability, ease of use, and wear-related concerns. A few concerns related to safety, loss, or appearance were raised similar to our study findings [24]. However, our findings differed from a study conducted on acceptability of GPS technology among patient/caregivers where the patient complained of the device's shape which they found too big [25]. The GPS size was more like a portable phone whereas in our study the GPS was very small and easy to carry that study participants liked. This implies that if a suitable GPS device is designed, it can be used by different populations in different settings.

Our findings demonstrate a willingness to use the GPS devices. The use of the GPS devices for tracking has taken a noticeable role in physical activity and exposure research [26]. This may help in tracing participants for future health research innovations.in fishing communities with high mobility rates. However, the designers may have to deal with expressed fears of devices being stolen or getting lost or damaged by water. Already women had several suggestions that could be considered during the re-design of the GPS devices to make them user friendly and appealing.

Our findings revealed that tracking devices like phones and GPS would not only be used to establish mobility patters but also trace women and other participants in future health research innovations.to improve retention. While both devices require charging at individual level, charging phones would be easier compared to GPS device that requires study staff to pick them up for charging and data download. Phones allow for dialogue on visit attendance and any other study related issues. However, basing on one of the studies conducted in fishing communities that compared physical tracing and phone use, phone use does provide room for leaving messages with other persons (family, workmates, and community members) to relay to the participant. This compromises confidentiality [13]. However, this can be solved by the use of GPS technology. Whereas the challenge of phone loss and unavailability due to lack of electricity at the time of reminders may affect tracing of participants in a study. It is true that mobile phone technology (text messages or audio call) allows flexibility in the time one can get in touch with participants (day or night) and regardless of their physical location as long as the mobile phone network is stable. Phones are known to have several disadvantages such as unreliable connectivity and lack of constant electricity for charging particularly in low resourced settings like countries in East Africa. Other challenges associated with use for mobility tracking are temporary and permanent change of phone ownership, loss of phones, and inability to comprehend a message especially text messages[27].

There continues to be a big debate in the level of male involvement in some of the health services women seek at health facilities [28]. Similarly, our findings indicate the need to involve

male spouses to authorize women's participation in health research activities. In settings where cultural traditions and gender norms support a more restricted decision-making role for women in general, there may be barriers of women participation in research without consultation of male partners. As such, sensitization of male partners is key to help them understand why women are participating in research.

This study was able to determine the feasibility and acceptability of using phones and tracking devices for tracking participants, in a HIV high-risk population that are potentially eligible for future clinical trials. Additionally, we conducted formal qualitative assessments on participants' perspectives of carrying GPS and phones. However, one of the limitations of this study we spotted was, while we gathered data from a number of sites along Lake Victoria, the study could not allow for direct comparison between sites to establish if there is noticeable differences in women's perspectives on GPS and phone technology. Conclusion.

Overall, the women in these fishing communities embraced the use of phones and GPS devices as a mobility tracking method and preferred using their own phones. However, they recommended that the community members be sensitized to avoid misunderstandings. Participants may be concerned that their partners or other associates will not understand the research goals so providing community wide information can help to dispel concerns. It was also recommended to change the colour and design of the GPS devices if it were to be used in future, to make them less obvious and avoid question about what they were. Thus phones and GPS devices can used in tracking movement patterns of mobile populations.

## Acknowledgments

The authors who also members of the Lake Victoria Consortium for Health Research (LVCHR) are very grateful to the Management of Uganda Virus Research Institute-International AIDS Vaccine Initiative HIV Vaccine Program, Kenya Medical Research Institute, Medical Research Council, Uganda Research Unit and; the Mwanza Intervention Trials Unit and Tanzania's National Institute for Medical Research and to the Leadership of the LVCHR for supporting the implementation of the study.

We are indebted to Professors: Elizabeth Bukusi affiliated to Research Care and Training Program (RCTP), Kenya Medical Research Institute, Kisumu, Kenya, Heiner Grosskurth from Mwanza Intervention Trials Unit (MITU), National Institute for Medical Research, Mwanza and Anatoli Kamali from IAVI, for the leadership provided to the LVCHR. Besides, we thank the stakeholders at community level in landing sites and at the Island that we worked with for their support, the study participants for providing the data; research assistants for collecting data, the Research and Ethics Committees of UVRI, KEMRI and MITU for reviewing the study; and the UVRI-IAVI, KEMRI, MITU Community Advisory Board (CAB) for their continuous advice and guidance throughout the implementation phase.

The contents of this manuscript are the responsibility of the authors and do not necessarily reflect the views of USAID or the US Government. We thank the reviewers for their insightful comments on the manuscript. We are greatly indebted to the field research team from UVRI-IAVI, MRC/UVRI and LSHTM, KEMRI and MITU for their tireless efforts in collecting data. This work was funded in part by IAVI and made possible by the support of many donors, including United States Agency for International Development (USAID). A list of IAVI donors is available at http://www.iavi.org.

## Author Contributions

**Conceptualization:** Matt A. Price, Saidi Kapiga, Pat Fast, Elizabeth Bukusi, Janet Seeley.

**Formal analysis:** Gertrude Nanyonjo, Zachary Kwena, Sarah Nakamanya, Elialilia Okello, Bertha Oketch.

**Funding acquisition:** Brenda Okech, Pat Fast.

**Investigation:** Saidi Kapiga.

**Methodology:** Gertrude Nanyonjo, Zachary Kwena, Matt A. Price, Elizabeth Bukusi.

**Project administration:** Janet Seeley.

**Supervision:** Gertrude Nanyonjo, Elialilia Okello, Bertha Oketch, Ubaldo M. Bahemuka, Ali Ssetaala, Brenda Okech, Saidi Kapiga, Elizabeth Bukusi, Janet Seeley.

**Writing – original draft:** Gertrude Nanyonjo.

**Writing – review & editing:** Zachary Kwena, Sarah Nakamanya, Elialilia Okello, Bertha Oketch, Ubaldo M. Bahemuka, Ali Ssetaala, Brenda Okech, Matt A. Price, Saidi Kapiga, Pat Fast, Elizabeth Bukusi, Janet Seeley.

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
