## [Decision Letter · Decision Letter 0]

2 Dec 2022

PONE-D-22-04440Finding Women in Fishing Communities around Lake Victoria: A  qualitative study on the use of phones and tracking devices.PLOS ONE

Dear Getrude Nanyonjo,

Thank you for submitting your manuscript to PLOS ONE. After careful consideration, we feel that it has merit but does not fully meet PLOS ONE’s publication criteria as it currently stands. Therefore, we invite you to submit a revised version of the manuscript that addresses the points raised during the review process.

ACADEMIC EDITOR:

Thank for a very useful study that can inform future innovative research and interventions in Kenya, and similar contexts. Three independent reviewers have taken time to provide a very thorough critique of your manuscript. They have provided very useful insights and ideas that can improve your final manuscript.

Pay attention to every comment they have sent. Address the following:

-Grammatical errors. Edit accordingly.

-The specific goal of this particular study and significant impact is broadly stated and keeps shifting. This is confusing. Kindly provide a focused version that runs throughout the manuscript.

-Confirm the number of study participants. How many participated in specific study activities? Be consistent throughout the manuscript

-Study design. The study design has elements of a longitudinal study, however, you report it as a cross-sectional study. Clarify the study design. Secondly, the study is more of an exploration, as opposed to ‘an assessment.’ Ensure your study design is therefore appropriately reported. Closely related to this observation, consider revising the title to match what is in the manuscript body better

-Provide a more comprehensive description of devices used in this study e.g. features

-Provide details about the study team and their qualifications

-Did you do any audio recording? Explain the illustrative quotes provided

-Ensure there is reporting of findings by differences in devices and other variables e.g. by age, ownership of a device, and sex…if at all men were involved! They are clearly missing in the methodology but mentioned in the findings chapter. Ensure illustrative quotes match the codes/themes provided

-Ensure the study is compared to the cited literature in the discussion section

-There should be no content discussed in the discussion section and ending remarks that was not provided in the findings. Align all sections of the manuscript.

-Expand study limitations e.g. exclusion of men from the study. Would their opinions have implications for use of this technology in this milieu? Reflect, explain, justify your position.

We look forward to receiving your revised manuscript.

Kind regards,

Violet Naanyu, PhD

Academic Editor

PLOS ONE

Journal Requirements:

4. One of the noted authors is a group or consortium [LVCHR study team]. In addition to naming the author group, please list the individual authors and affiliations within this group in the acknowledgments section of your manuscript. Please also indicate clearly a lead author for this group along with a contact email address.

5**. **Please ensure that you refer to Figures 1 & 2 in your text as, if accepted, production will need this reference to link the reader to the figure.

Additional Editor Comments (if provided):

None added here. See comments provided above.

Reviewers' comments:

Reviewer's Responses to Questions

**Comments to the Author**

1. Is the manuscript technically sound, and do the data support the conclusions?

Reviewer #1: Yes

Reviewer #2: Partly

Reviewer #3: Partly

2. Has the statistical analysis been performed appropriately and rigorously? 

Reviewer #1: N/A

Reviewer #2: N/A

Reviewer #3: N/A

3. Have the authors made all data underlying the findings in their manuscript fully available?

Reviewer #1: Yes

Reviewer #2: Yes

Reviewer #3: No

4. Is the manuscript presented in an intelligible fashion and written in standard English?

Reviewer #1: Yes

Reviewer #2: Yes

Reviewer #3: Yes

5. Review Comments to the Author

Reviewer #1: Congratulations on undertaking an innovative important study. Your paper, including the introduction, methods, findings and discussion are all methodologically sound, logical and clearly articulated. This paper was a pleasure to read and the findings will be highly applicable to future studies and interventions in your region.

There were a few grammatical errors. I have made suggested corrections in the comments below that do not request a re-write by the authors.

Line 265: However, majority of the women liked the size of the GPS much as they made suggestions for consideration to further reduce to a size of a Subscriber Identity module (SIM) card or ring. This sentence does not make sense and needs to be re-written.

Line 374: a study conducted among patient/caregivers where the patient focused

Line 385: While both devices require charging at the individual level,

Line 388: However, basing on one of the studies conducted in fishing communities that compared physical tracing and phone use, phone use does not provide room for leaving messages with other persons 23 (family, workmates, community members) to relay to the participant in cases of no contact and ensure confidentiality (14). NEEDS RE-WORDING.

Line 396: lack of constant electricity for charging

Line 412: Overall, the women in these fishing communities embraced the use of phones and GPS devices as a mobility tracking method. Their preference was for using their own phones if community sensitization had occurred and had a further recommendation of a change in colour and design of the GPS devices if it were to be used. Thus phones and GPS devices can used in tracking movement patterns of mobile populations.

Reviewer #2: Line 34 -36 reads: “Sixty-four 35 women participated: 24 were given a study phone; 24 were asked to allow tracking The following comments indicate why I think the manuscript is partly technically sound. The comments call for minor but very significant revisions.

of their own 36 phones and 16 were provided with a portable GPS device to understand what is most preferred.” But

Table 1 (line 181-182 ) reads 48 participants. What happened to the 16 participants?

It would appear that there were design and ethics issues which could have been avoided with more robust community engagement in conceptualization and design of the study or/and by mainstreaming gender in this study. What was the justification for engaging women only in this study? In other words, why were men excluded from this study? Would this not imply that in the fishing communities, women are the spreaders of HIV? As the findings indicate, did the study not expose women to risk of domestic violence? And were the women who participated in this study HIV positive? Were they all married or having male sexual partners now that they indicate that their partners would object to their carrying devices and would question where they got money from to buy the devices.

What is the goal of the study? Sometimes it is stated as “to inform future research innovations and interventions.’ (line 308). Does this mean all kinds of research innovations and interventions or health research innovations and interventions or both health research and health care innovations and innovations? There is need for clarity on this. Mixing health care with health research is particularly dangerous because it could lead to therapeutic misconception. Moreover, would the results be the same if the study was on health care only or health research only or both? I would imagine so. Asking me to carry a device for purposes of improving research is different from asking me to carry a device for purposes of improving health care. In other areas, there is reference to clinical research and clinical trials and public health and yet in other areas the focus is limited to informing methods, to informing larger intervention studies, to informing design of future studies etc. It is important to set the bounds of this study clearly.

The study design is not that of an assessment but of an exploratory cross sectional descriptive study. The study explores and describes the experiences and perceptions of women in use of phones and GPS to track mobility.

Line 156 suggests that training on ‘ethical issues that would be encountered during field work’ was conducted. It would not have been possible to know the ethical issues that would arise so training would only have been on ethical considerations. Ordinarily, no researcher is definite on the ethical issues that they are likely to find in the study.

Reviewer #3: Finding women in fishing communities around Lake Victoria: A qualitative study on the use of phones and tracking devises.

This is an interesting paper with potential to inform projects or studies enrolling highly mobile participants – to achieve both high retention and improve validity of the study findings and or impact of the projects. Although the paper provides useful information on experiences and perceptions including feasibility and acceptability to use phones and tracking devices in studies, the manuscript requires major improvement before its consideration for publication.

Title: The title is well written and provides information on the population and the study setting as well as the design employed. I would suggest improving the title to identify with the objective of the study “Finding women in fishing communities around Lake Victoria: feasibility and acceptability of using phones and tracking devises”.

Abstract: The abstract is well structured and concise. Can revise the first sentence in methods section to read – ‘A formative qualitative methods study…’. It is necessary to clarify how many participants were enrolled in this study – Is it 64, 60 or 48?

Introduction: Generally, the introduction section is focused to what the manuscript is communicating. Provides a background to why the study is being conducted including the set-out objectives.

Methods: This section requires clarity. There are various sections that one is left wondering what was done and how was it done. First, the design is confusing, there seems to have been enrolment and follow up of participants either through FGDs or in-depth interviews for four months asking information related to their wellbeing, movements and experiences using the devices. However, the authors mention that this was a cross-sectional study while this seem to have had data collected over time with possibilities of experiences changing over time. It might be helpful to describe the type of phone provided by the study to participants as this could likely have had an effect on the findings of the study.

Data collection and management – the reader may benefit understanding who the research team that conducted the interviews and FGD was, what was their qualification including experience in conducting qualitative interviews. The fact that notes were taken during the interview and expanded after the interview meant that the quotes being provided at not verbatim, and that needs to be clarified. In line 158-177 provides the number of participants enrolled (18 women, 6 from each country randomized to simple study phone; another similar group – here I assume 18 to use their own mobile phone or a mobile phone they have access to; 24 women were given a GPS device) all these gives a total sample of 60 participants. Who was engaged in FGD and how many FGD were conducted in total? Were the same questions asked in IDIs and FGDs? How were the 4 participants in each site selected for IDIs? How was data saturation determined, if it was?

Table 1: This table seems to provide a total of 48 participants in the study, different from what is described above. There is a word missing between who and GPS on the second row (category column) “women who GPS while in the community”

Results: The first sentence reads, “We enrolled 48 women (24 per country) – the expected number here would be 64 (24*3 countries). The thematic area (e) experiences of those who carried GPS and phone is vague. Are these experiences unique from what is grouped in a-d? Beyond what is summarized on the table, concerns about male involvement is provided as a theme/group.

Table 2 – it might be useful to categorize experiences of those who used their own phones vs. those who used the study phone (may necessitate providing three columns with phone users broken down into two)

Line 259-262 provides a quote from Youn men FGD, Kenya. In the manuscript all the study participants were women, and not exactly sure how males were involved in this study. This is supported by a statement in line 347-349 that men complained of not being informed as they did not have time to attend meeting that were organized at the beginning of the study. All these procedures including inclusion of men in the study is missing in the methods section.

In various instances, the descriptions provided do not match the quotes that exemplifies what is being described. The description on fears, worries and uncertainties around the device – line 296-297 describes an anticipated worry. At what point is the participant providing this description? After receipt of the device or before? It is my understanding that the participant would narrate experiences using a phone or gadget for tracking. Were they being asked a ‘what if’ question to necessitate the response received? In other instances, for example line 339-340 provides a description that is non-related to concerns about male involvement as a theme.

Line 316 provides a quote from Young women in Uganda – were participants in this study grouped into young and old? What were the age categories? Provide more details in the methods to help with this understanding.

The quotes seem to be quoted verbatim…. Not exactly sure how these were captured in a scenario where study team was taking notes and later expanding the notes taken ---- amounting to paraphrasing the participant’s verbal words.

Discussion: This section is fairly done. It might benefit from better situating this current study within the available literature. There are studies being compared to the current study, without contextualizing these differences. For example, line 373-375, states the difference from this study with one that was conducted among the patient/caregivers on the ‘shape’ that was seen to be ‘big’. It is unclear if the current study used the same/similar GPS device as the Alzheimer’s study

Line 386 mentions GPS device requiring study staff to pick them up for charging and data download. This information is not provided in the methods section. Line 398 provides information on comprehending text messages – were the participants sent messages? Were they meant to read and respond to them? In the methods, the authors mention calling participants on specific time intervals but not text messaging is mentioned to warrant this.

Line 404-406, discussing the importance of sensitizing men in women’s participation in research is too general and not informed by this study. This study focused on a device or provision of a phone, something that identified participants in the study among their male partners. This may not be true should a study require say, one-time information related to women without follow up information.

Strengths and limitations to this study is too vague. How was this study able to ‘evaluate’ the feasibility? Rewrite this section.

The last paragraph before acknowledgement I assume is the conclusion and should be titled as such. It reads ok although it needs copyediting in some areas …with preference of using their own phones if community is sensitized? Currently reads sensitization.

Please include COREQ checklist alongside the revised copy of the manuscript

6. PLOS authors have the option to publish the peer review history of their article (what does this mean?). If published, this will include your full peer review and any attached files.

Reviewer #1: No

Reviewer #2: No

Reviewer #3: No

---

## [Author Response · Author response to Decision Letter 0]

18 Jan 2023

Reviewer #1: Congratulations on undertaking an innovative important study. Your paper, including the introduction, methods, findings and discussion are all methodologically sound, logical and clearly articulated. This paper was a pleasure to read and the findings will be highly applicable to future studies and interventions in your region. There were a few grammatical errors. I have made suggested corrections in the comments below that do not request a re-write by the authors.

Response: Thank you

Line 265: However, majority of the women liked the size of the GPS much as they made suggestions for consideration to further reduce to a size of a Subscriber Identity module (SIM) card or ring. This sentence does not make sense and needs to be re-written.

Response: Thank you for the observation, I have re-written the section line 318-320. It reads “However, majority of the women suggested the need to reduce the GPS size to a size of a Subscriber Identity module (SIM) card or finger ring”

Line 374: a study conducted among patient/caregivers where the patient focused

Response: Thank you, the correction has been considered line 432

Line 385: While both devices require charging at the individual level,

Response: Thank you, the correction has been considered line 445

Line 388: However, basing on one of the studies conducted in fishing communities that compared physical tracing and phone use, phone use does not provide room for leaving messages with other persons 23 (family, workmates, community members) to relay to the participant in cases of no contact and ensure confidentiality (14). NEEDS RE-WORDING.

Response: Thank you for the suggestion, the sentence is has been written line448-452. It reads “However, basing on one of the studies conducted in fishing communities that compared physical tracing and phone use, phone use does provide room for leaving messages with other persons (family, workmates, and community members) to relay to the participant. This compromises confidentiality (14). However, this can be solved by the use of GPS technology”

Line 396: lack of constant electricity for charging

Response: Thank you, the correction has been considered line 457

Line 412: Overall, the women in these fishing communities embraced the use of phones and GPS devices as a mobility tracking method. Their preference was for using their own phones if community sensitization had occurred and had a further recommendation of a change in colour and design of the GPS devices if it were to be used. Thus phones and GPS devices can used in tracking movement patterns of mobile populations.

Response: Thank you. The statement has been revised line 478-484

Reviewer #2: Line 34 -36 reads: “Sixty-four 35 women participated: 24 were given a study phone; 24 were asked to allow tracking. The following comments indicate why I think the manuscript is partly technically sound. The comments call for minor but very significant revisions

of their own 36 phones and 16 were provided with a portable GPS device to understand what is most preferred.” But Table 1 (line 181-182 ) reads 48 participants. What happened to the 16 participants? It would appear that there were design and ethics issues which could have been avoided with more robust community engagement in conceptualization and design of the study or/and by mainstreaming gender in this study. What was the justification for engaging women only in this study? In other words, why were men excluded from this study? Would this not imply that in the fishing communities, women are the spreaders of HIV? As the findings indicate, did the study not expose women to risk of domestic violence? And were the women who participated in this study HIV positive? Were they all married or having male sexual partners now that they indicate that their partners would object to their carrying devices and would question where they got money from to buy the devices

Response: Thank you for the observation, the sample size has been corrected and the table has revised 189-201and line 228. It reads “ Basing on the suggestions from the community focus group discussions, the identified women were invited to participate in the study. One group of 24 women, 6 from each implementing research institute were randomized to simple study mobile phone provided by the research institute that included credit for telephone use and another similar group of 24 were asked to use their own mobile phones or a mobile phone they have access to and given roughly USD 1.5 worth of credit for telephone use. Each woman from the groups that received/or used their own phones was contacted by phone after three, six and nine days by the research team and asked about any movements made outside the community they live in over the past 3 days. In addition, face-to-face IDIs were conducted monthly. In addition, a total of 16 women, 4 from each implementing site in Kenya, Tanzania and Uganda were given a small portable GPS device to carry for 4 months (see Table 1) line 210. Women were instructed on how to hand the device and how often the research would collect the devices to download data and charge the GPS. Based on the choice made during community discussions, four women from each site were identified to be invited to take part in the study

Why women only?

Response: Thank you for the concern. studies conducted in fishing communities of Lake Victoria in East Africa have reported high HIV prevalence and incidence, with more than half of the women infected with HIV in some age groups reference are indicated ( 2 and 3) line 521-522. Many women in fishing communities are very mobile; they change places of work and sexual partners frequently, which results in increased risk of acquisition of HIV and other sexually transmitted infections as indicated in the introduction but viewed as the only spreaders.

The methods section design line 122-125 and Study setting 153-156 indicates how men were involved.

What is the goal of the study? Sometimes it is stated as “to inform future research innovations and interventions.’ (line 308). Does this mean all kinds of research innovations and interventions or health research innovations and interventions or both health research and health care innovations and innovations? There is need for clarity on this. Mixing health care with health research is particularly dangerous because it could lead to therapeutic misconception. Moreover, would the results be the same if the study was on health care only or health research only or both? I would imagine so. Asking me to carry a device for purposes of improving research is different from asking me to carry a device for purposes of improving health care. In other areas, there is reference to clinical research and clinical trials and public health and yet in other areas the focus is limited to informing methods, to informing larger intervention studies, to informing design of future studies etc. It is important to set the bounds of this study clearly.

Response: Thank you. The goal to inform future health research innovations. Revisions have been made on line 32, 168-169 and 172-173, 440 and 446

The study design is not that of an assessment but of an exploratory cross sectional descriptive study. The study explores and describes the experiences and perceptions of women in use of phones and GPS to track mobility.

Response: Thank you for this observation, this was a multi-site formative qualitative study line 116

Line 156 suggests that training on ‘ethical issues that would be encountered during field work’ was conducted. It would not have been possible to know the ethical issues that would arise so training would only have been on ethical considerations. Ordinarily, no researcher is definite on the ethical issues that they are likely to find in the study.

Response: Thank you for the guidance, this has been corrected line 186

Reviewer #3: Finding women in fishing communities around Lake Victoria: A qualitative study on the use of phones and tracking devises. 

This is an interesting paper with potential to inform projects or studies enrolling highly mobile participants – to achieve both high retention and improve validity of the study findings and or impact of the projects. Although the paper provides useful information on experiences and perceptions including feasibility and acceptability to use phones and tracking devices in studies, the manuscript requires major improvement before its consideration for publication. 

Title: The title is well written and provides information on the population and the study setting as well as the design employed. I would suggest improving the title to identify with the objective of the study “Finding women in fishing communities around Lake Victoria: feasibility and acceptability of using phones and tracking devises”. 

Response: Thank you for the guidance. The tittle has been revised as suggested “Finding Women in Fishing Communities around Lake Victoria: “feasibility and acceptability of using phones and tracking devices”. See line 1-3

Abstract: The abstract is well structured and concise. Can revise the first sentence in methods section to read – ‘A formative qualitative methods study…’. It is necessary to clarify how many participants were enrolled in this study – Is it 64, 60 or 48? 

Response: The first sentence has been revised, line 34. The total number of study participants who received the devices were 64 (48 participants in the phone arm and 16 the GPS arm). Changes have been made. See line 37

Introduction: Generally, the introduction section is focused to what the manuscript is communicating. Provides a background to why the study is being conducted including the set-out objectives. 

Response: Thank you.

Methods: This section requires clarity. There are various sections that one is left wondering what was done and how was it done. First, the design is confusing, there seems to have been enrolment and follow up of participants either through FGDs or in-depth interviews for four months asking information related to their wellbeing, movements and experiences using the devices. However, the authors mention that this was a cross-sectional study while this seem to have had data collected over time with possibilities of experiences changing over time. It might be helpful to describe the type of phone provided by the study to participants as this could likely have had an effect on the findings of the study. 

Responses: Thank you for this observation. This was a multisite formative qualitative study. We have made revisions on line 115. The phones were basic inexpensive mobile phones loaded with weekly credit of roughly 1.5 US dollars line 127-128

Data collection and management – the reader may benefit understanding who the research team that conducted the interviews and FGD was, what was their qualification including experience in conducting qualitative interviews. The fact that notes were taken during the interview and expanded after the interview meant that the quotes being provided at not verbatim, and that needs to be clarified. In line 158-177 provides the number of participants enrolled (18 women, 6 from each country randomized to simple study phone; another similar group – here I assume 18 to use their own mobile phone or a mobile phone they have access to; 24 women were given a GPS device) all these gives a total sample of 60 participants. Who was engaged in FGD and how many FGD were conducted in total? Were the same questions asked in IDIs and FGDs? How were the 4 participants in each site selected for IDIs? How was data saturation determined, if it was?

Response: We have included in the methods section under study setting line 139-145 that: This was a multi-site qualitative study conducted by investigators from the partners of the Lake Victoria Consortium for Health Research (LVCHR). LVCHR members include: i) Kenya Medical Research Institute (KEMRI); ii) Mwanza Intervention Trials Unit (MITU) in Tanzania; iii) International AIDS Vaccine Initiative unit within the Uganda Virus Institute (UVRI/IAVI); and iv) the Medical Research Council/Uganda Virus Research Institute Uganda Research Unit & London School of Hygiene and Tropical Medicine (MRC/UVRI & LSHTM) Uganda Research Unit. 

We have still indicated under data collection on line 175-177 that; ‘The Consortia leadership conceived and designed the study. Each site had one senior social science/behavioral researcher who coordinated activities of the study. These comprised of protocol training, supervision of field teams in data collection and management activities. 

There were four field researchers at each site, both male and female field researchers of various ages (ranging between 24 to 40), who were selected from the research teams at each of the participating institutions based on their experience and training. The field researchers had enough experience to conduct research in the study settings.’ We have included a flowchart (Fig 2) as an attachment indicating the structure of the research teams. Line 178-182

Fig 3: Flowchart on structure of the research teams

Besides, we have corrected the sample size line 186-199; One group of 24 women, 6 from each implementing research institute were randomized to simple study mobile phone provided by the research institute that included credit for telephone use and another similar group of 24 were asked to use their own mobile phones or a mobile phone they have access to and given roughly USD 1.5 worth of credit for telephone use. Each woman from the groups that received/or used their own phones was contacted by phone after three, six and nine days by the research team and asked about any movements made outside the community they live in over the past 3 days. In addition, face-to-face IDIs were conducted monthly. In addition, a total of 16 women, 4 from each implementing site in Kenya, Tanzania and Uganda were given a small portable GPS device to carry for 4 months (see Table 1) line 225. Women were instructed on how to hand the device and how often the research would collect the devices to download data and charge the GPS.

Based on the choice made during community discussions, four women from each site were identified to be invited to take part in the study.

Table 1: This table seems to provide a total of 48 participants in the study, different from what is described above. There is a word missing between who and GPS on the second row (category column) “women who GPS while in the community”

Response: The numbers in the table have been reconciled. The total number of participants who carried the devices were 64. See line 225. The missing word has been added

Results: The first sentence reads, “We enrolled 48 women (24 per country) – the expected number here would be 64 (24*3 countries). The thematic area (e) experiences of those who carried GPS and phone is vague. Are these experiences unique from what is grouped in a-d? Beyond what is summarized on the table, concerns about male involvement is provided as a theme/group. 

Response: Thank you for this suggestion. The total number of study participants has been reconciled. We did however, change the numbers because we have 16 participants in four sites, so we enrolled 64 participants for both arms. See line 241

Table 2 – it might be useful to categorize experiences of those who used their own phones vs. those who used the study phone (may necessitate providing three columns with phone users broken down into two) 

Line 259-262 provides a quote from Young men FGD, Kenya. In the manuscript all the study participants were women, and not exactly sure how males were involved in this study. This is supported by a statement in line 347-349 that men complained of not being informed as they did not have time to attend meeting that were organized at the beginning of the study. All these procedures including inclusion of men in the study is missing in the methods section. 

Response: The table has been broken into two for the column of phone. We have included a statement on how males were involved in the study under study setting line 152-154“. Although our study population comprised of women residents and /or working in fishing communities.

Men were also involved during entry meetings and Focus group community discussions where shared their views about the feasibility of using GPS and phone devices, identifying and how to approach women to participate in the study. A statement has been included in the methods section-sampling procedure line 157-158

In various instances, the descriptions provided do not match the quotes that exemplifies what is being described. The description on fears, worries and uncertainties around the device – line 296-297 describes an anticipated worry. At what point is the participant providing this description? After receipt of the device or before? It is my understanding that the participant would narrate experiences using a phone or gadget for tracking. Were they being asked a ‘what if’ question to necessitate the response received? In other instances, for example line 339-340 provides a description that is non-related to concerns about male involvement as a theme. 

Response: This was before they carried the devices. In response to description, this has been revised under results line 327-331 “Other felt that their sexual partners would question why they are carrying such devices. Some were scared of losing the GPS. On the other hand, the phone users in both arms presented issues of inconsistent power supply, network challenges. Those who were provided with institutional phones expressed the burden of carrying two phones.

Line 316 provides a quote from Young women in Uganda – were participants in this study grouped into young and old? What were the age categories? Provide more details in the methods to help with this understanding. 

Response: Thank for this observation, Yes, those who participated in the community FGDs were grouped according to young women, elder women, young men, elder men. See line 121-124 “ The discussions were held with residents of FCs (young women and men 18-25 years of age, older women, and older men 26 years+) separately to explore acceptability of using GPS devices and phones and how best individuals could be identified and approached to participate in carrying the devices”.

The quotes seem to be quoted verbatim…. Not exactly sure how these were captured in a scenario where study team was taking notes and later expanding the notes taken ---- amounting to paraphrasing the participant’s verbal words. 

Response: A statement on how voices were documented as quotes is included line 230-231under data management and analysis “An effort was made to maintain participants’ voices as quotes on interesting information that came up during the interview”

Discussion: This section is fairly done. It might benefit from better situating this current study within the available literature. There are studies being compared to the current study, without contextualizing these differences. For example line 373-375, states the difference from this study with one that was conducted among the patient/caregivers on the ‘shape’ that was seen to be ‘big’. It is unclear if the current study used the same/similar GPS device as the Alzheimer’s study

Response: The sentence has been revised line 428-432 “However, our findings differed from a study conducted on acceptability of GPS technology among patient/caregiver where the patient complained of the device's shape which they found too big (26). The GPS size was more like a portable phone whereas ours was very small and easy to carry that study participants liked”

Line 386 mentions GPS device requiring study staff to pick them up for charging and data download. This information is not provided in the methods section. Line 398 provides information on comprehending text messages – were the participants sent messages? Were they meant to read and respond to them? In the methods, the authors mention calling participants on specific time intervals but not text messaging is mentioned to warrant this. 

Response: Thank you for this observation, a statement has been included under data collection and management line 208-209 “Women were instructed on how to hand the device and how often the research would collect the devices on a monthly to download data and charge the GPS. Women were met in a specified place on a monthly basis for 4 months and asked about their wellbeing, movements within the preceding 30 days and about their experience with carrying the device”

Line 404-406, discussing the importance of sensitizing men in women’s participation in research is too general and not informed by this study. This study focused on a device or provision of a phone, something that identified participants in the study among their male partners. This may not be true should a study require say, one-time information related to women without follow up information. 

Response: Thank you, however, during the FGD, women clearly pointed out that it is important to sensitize men (sexual partners). Basing on our experience, it is a general issue and relevant that may require attention for future studies.

Strengths and limitations to this study is too vague. How was this study able to ‘evaluate’ the feasibility? Rewrite this section. 

Response: Thank you, this has been revised line 462-463. It reads “This study was able to determine the feasibility and acceptability of using phones and tracking devices for tracking participants, in a HIV high-risk population that are potentially eligible for future clinical trials”

Added a limitation line 467-470 which states “However, one of the limitations of this study we spotted was, while we gathered data from a number of sites along Lake Victoria, the study could not allow for direct comparison between sites to establish if there is noticeable differences in women’s perspectives on GPS and phone technology”

The last paragraph before acknowledgement I assume is the conclusion and should be titled as such. It reads ok although it needs copyediting in some areas …with preference of using their own phones if community is sensitized? Currently reads sensitization. 

Response: Much appreciated for your observation, the sentence has been revised line 468-480. It reads “Overall, the women in these fishing communities embraced the use of phones and GPS devices as a mobility tracking method and preferred using their own phones. However, they recommended that the community members be sensitized to avoid misunderstandings. Participants may be concerned that their partners or other associates will not understand the research goals so providing community wide information can help to dispel concerns. It was also recommended to change the colour and design of the GPS devices if it were to be used in future, to make them less obvious and avoid question about what they were. Thus phones and GPS devices can used in tracking movement patterns of mobile populations. ”

---

## [Decision Letter · Decision Letter 1]

5 Apr 2023

PONE-D-22-04440R1Finding Women in Fishing Communities around Lake Victoria: “feasibility and acceptability of using phones and tracking devices”.PLOS ONE

Dear Dr. Getrude Nanyonjo,

Thank you for submitting your manuscript to PLOS ONE. After careful consideration, we feel that it has merit but does not fully meet PLOS ONE’s publication criteria as it currently stands. Therefore, we invite you to submit a revised version of the manuscript that addresses the points raised during the review process.

We look forward to receiving your revised manuscript.

Kind regards,

Violet Naanyu, PhD

Academic Editor

PLOS ONE

Journal Requirements:

Additional Editor Comments:

Thank you for the revised version of your manuscript. There are a few pending areas in the methodology that require clarity especially the inclusion of male participants, consequent data analysis & reporting as expansively flagged by one of the reviewers. I had already raised these queries before.

I look forward to your response on this matter so that we can move to the next step. Thank you.

Reviewers' comments:

Reviewer's Responses to Questions

**Comments to the Author**

1. If the authors have adequately addressed your comments raised in a previous round of review and you feel that this manuscript is now acceptable for publication, you may indicate that here to bypass the “Comments to the Author” section, enter your conflict of interest statement in the “Confidential to Editor” section, and submit your "Accept" recommendation.

Reviewer #1: All comments have been addressed

Reviewer #2: All comments have been addressed

Reviewer #3: (No Response)

2. Is the manuscript technically sound, and do the data support the conclusions?

Reviewer #1: Yes

Reviewer #2: Yes

Reviewer #3: Yes

3. Has the statistical analysis been performed appropriately and rigorously? 

Reviewer #1: N/A

Reviewer #2: N/A

Reviewer #3: N/A

4. Have the authors made all data underlying the findings in their manuscript fully available?

Reviewer #1: Yes

Reviewer #2: Yes

Reviewer #3: Yes

5. Is the manuscript presented in an intelligible fashion and written in standard English?

Reviewer #1: Yes

Reviewer #2: Yes

Reviewer #3: (No Response)

6. Review Comments to the Author

Reviewer #1: Thank you for responding to my review comments.

I have no competing conflicts of interest in reviewing this paper.

Reviewer #2: Everything is fine but the authors still need to confirm the numbers of the participants across the various sites.

Reviewer #3: Detailed comments are provided in the attachment

I still find the summary of the methods section in the abstract, main body methods section (line 112-122), sampling procedure (line 152-158) of the manuscript confusing due conflicting information provided.

I am left wondering how one would interview a male to document experiences of women who carried either phones or GPS devises?

Beyond this, I have not read anywhere who (participant characteristics), how they were selected, how many FGD conducted among them, what was asked of them among other information relevant to this group. Your sampling procedure line 152-158 does not provide this.

I get a sense that the gatekeepers were only approached to obtain support and buy in and that only women were recruited for the study and interviewed (IDI and or FGD) to explore their perceptions and experiences of using GPS devise and phone for tracking.

Your results section line 247-249 supports what I state above. Seems only women were recruited for this study or likely a sub-study of the main study.

If this is a sub-analysis from a larger study, I would understand this confusion and there needs to be a delineation to every manuscript and its focus. If you are only focusing on women’s perception on use of the phones and GPS devices for tracking – including the pilot study to assess feasibility, then present this sub analysis only. As your male participants are not represented in the study population (64 participants), neither are they selected in the feasibility study, then remove their reference in the study.

The results section presenting the male participants in this study might be removed should the above be considered and similarly the discussion section referring to it.

Why was this study limited to direct comparison of data between sites? I believe the data was collected and segregated per site and as such if there were concerns raised by communities in one site compared to another, these could be shared. Otherwise, if these concerns cut across the three sites/countries – then can certainly say we did not find any differing views/perceptions/experiences of women in these communities hence the presentation of the results and discussion as such.

7. PLOS authors have the option to publish the peer review history of their article (what does this mean?). If published, this will include your full peer review and any attached files.

Reviewer #1: No

Reviewer #2: No

Reviewer #3: No

---

## [Author Response · Author response to Decision Letter 1]

15 Jun 2023

RE: Response to reviewers’ comments for manuscript number: PONE-D-22-04440R1

We appreciate the opportunity to respond to the comments and we are grateful to both the Editor and Reviewers who took the time to review our manuscript and provided helpful comments to improve the manuscript. All the comments have been addressed point by point as indicated below. Where we include line numbers, we refer to the track change version of the paper. We hope that the reviewers find our revisions adequate.

Editor(s)' Comments:

I still find the summary of the methods section in the abstract, main body methods section (line 112-122), sampling procedure (line 152-158) of the manuscript confusing due conflicting information provided. 

Response: Thank you for the observation. This has been revised see line 35-41. It reads “Sixty-four (64) women participated in the study (16 per fishing community). Twenty-four (24) participants were given a study phone; 24 were asked to use their own phones and 16 were provided with a portable GPS device to understand what is most preferred. Women were interviewed about their experiences and recommendations on carrying GPS devices or phones. Twenty four (24) Focus Group Discussions with 8-12 participants were conducted, 4 FGDS in each fishing community with community members to generate data on community perceptions regarding GPS devices and phones acceptability among women. Data were analysed thematically and compared across sites/countries”.

Line 112-122 reads “This study used community focus group discussions (FGDs) and in-depth interviews (IDIs) to document the perceptions, and experiences of women in fishing communities who carried either a GPS device or a phone. Community FGDs were considered appropriate for generating data on community perceptions regarding GPS devices and phones acceptability among women. The discussions were held with residents of FCs (young women and men 18-25 years of age, older women, and older men 26 years+) separately to explore acceptability of using GPS devices and phones and how best individuals could be identified and approached to participate in carrying the devices. IDIs were considered appropriate for generating data on personal lived experiences of those who actually carried the GPS devices, their own phones and study phones handed out by the researchers”

Yellow highlight - I am left wondering how one would interview a male to document experiences of women who carried either phones or GPS devises? 

Response: The methods sections has been revised 114-123 reads “This study used in-depth interviews (IDIs) to document the perceptions, and experiences of women in fishing communities who carried either a GPS device or a phone. Besides, community FGDs were considered appropriate for generating data on community perceptions regarding GPS devices and phones acceptability among women. The discussions were held with residents of FCs (young women and men 18-25 years of age, older women, and older men 26 years+) separately to explore their views on the acceptability of using GPS devices and phones by women. During the discussions, participants were also asked to suggest how best individuals could be identified and approached to participate in carrying the devices”

The green highlight – Beyond this, I have not read anywhere who (participant characteristics), how they were selected, how many FGD conducted among them, what was asked of them among other information relevant to this group. 

Response: Very sorry, we had missed including the participant characteristics. This has been resolved. See Table 1; Participant characteristics

Concerning how participants were selected, the number of FGD and what was asked, this has been clearly stated line 118- 123 “Besides, community FGDs were considered appropriate for generating data on community perceptions regarding GPS devices and phones acceptability among women. The discussions were held with residents of FCs (young women and men 18-25 years of age, older women, and older men 26 years+) separately to explore their views on the acceptability of using GPS devices and phones by women. During the discussions, participants were also asked to suggest how best individuals could be identified and approached to participate in carrying the devices”. 

Then 186-200 “We organized and conducted to 6 introductory meetings with community gate keepers and the community members in each of the six fishing communities. Each meeting consisted of 8–15 members who had lived in the fishing communities for at least 1 year. Those who participated in the introductory sessions were nominated by local leaders in the communities. During community meeting, contact persons were identifies to support in mobilizing individuals to part in the FGDs and those to carry GPS device and phones. We conducted 4 Focus Group Discussions in each of the fishing community to explore views from different groups of people on the acceptability of using GPS devices and phones by women, how communities would think about GPS device and how to identify the women who are mobile to participate in carrying GPS device. Group discussion participants were identified in two ways: (1) selected by community local leaders and, (2) referrals from introductory meetings of people who were knowledgeable on the subject of discussion. The groups were divided into four categories based on gender and age: young women (18–25), older women (25+), young men (18–25) and older men (25+). Basing on the suggestions from the community meetings, and FGDs, the contact persons identified during community meetings supported the process of mobilizing women to carry GPS and phones”.

Table 2: Participant Characteristics

Variables MRC/UVRI and LSHTM UVRI-IAVI MITU KEMRI Total

 Phone users GPS users Phone users GPS users Phone users GPS Users Phone users GPS users N = 64

Sex 

Male 0 0 0 0 0 0

Female 12 4 12 4 12 4 12 4 64

Age (years) 

18–24 5 2 3 1 2 2 2 1 18

25–34 3 1 5 2 6 1 5 2 25

35+ 4 1 4 1 4 1 5 1 21

Education 

None 1 0 0 0 0 0 * * 1

Primary 5 3 8 2 6 2 * * 26

Secondary 4 1 2 2 4 2 * * 15

Tertiary 2 0 2 0 2 0 * * 6

Occupation 

Fish related activities 4 1 3 2 5 1 2 2 20

Trader 1 1 2 1 2 5 1 13

Sex/bar worker 4 2 5 1 3 2 2 1 20

Other 3 0 2 0 2 1 3 0 11

Marital status 

Married/cohabiting 8 2 6 2 5 1 5 2 31

Separated/single 4 2 6 2 7 3 6 2 32

Widowed 0 0 0 0 0 0 1 0 1

* Data not captured 

FGD-Not collected 

Your sampling procedure line 152-158 reads 

“The process of identifying participants started by conducting community entry meetings where we met the gatekeepers (both male and female) to obtain their support and buy-in. During community entry meetings, local leaders were requested to help in identifying women aged 18-45 years willing to carry GPS or phones and to help identify private and confidential meeting venues as well as mobilize potential participants. The study team screened the identified potential participants for uncoerced willingness to participate in the study as well as eligibility based on age above 18, at least 6 months residence in the community, own a phone and frequency of mobility.”

In the highlighted section I get a sense that the gatekeepers were only approached to obtain support and buy in and that only women were recruited for the study and interviewed (IDI and or FGD) to explore their perceptions and experiences of using GPS devise and phone for tracking. 

Your results section line 247-249 supports what I state above. Seems only women were recruited for this study or likely a sub-study of the main study. 

“We enrolled 64 women (16 per site) between 18-45 years of age in fishing communities of Kenya, Uganda and Tanzania and conducted community FGD with different groups of people to seek their views on the use of the two devices.”

If this is a sub-analysis from a larger study, I would understand this confusion and there needs to be a delineation to every manuscript and its focus. If you are only focusing on women’s perception on use of the phones and GPS devices for tracking – including the pilot study to assess feasibility, then present this sub analysis only. As your male participants are not represented in the study population (64 participants), neither are they selected in the feasibility study, then remove their reference in the study. 

The results section presenting the male participants in this study might be removed should the above be considered and similarly the discussion section referring to it. 

Response: Different methods were employed. We first introduced the study through community meetings, held FGDs with community leaders/gate keepers and community members and later conducted IDI with only women who carried GPS and phones but at the same time had FGDs with community leaders. This has been clarified under the methods section

Why was this study limited to direct comparison of data between sites? I believe the data was collected and segregated per site and as such if there were concerns raised by communities in one site compared to another, these could be shared. Otherwise, if these concerns cut across the three sites/countries – then can certainly say we did not find any differing views/perceptions/experiences of women in these communities hence the presentation of the results and discussion as such. 

Response: Thank you so much, true we did not find any differing views /perceptions/experiences in the different communities.

---

## [Decision Letter · Decision Letter 2]

5 Jul 2023

PONE-D-22-04440R2Finding Women in Fishing Communities around Lake Victoria: “feasibility and acceptability of using phones and tracking devices”.PLOS ONE

Dear Dr. Nanyonjo,

Thank you for submitting your manuscript to PLOS ONE. After careful consideration, we feel that it has merit but does not fully meet PLOS ONE’s publication criteria as it currently stands. Therefore, we invite you to submit a revised version of the manuscript that addresses the points raised during the review process.

We look forward to receiving your revised manuscript.

Kind regards,

Jackie Sammonds

Staff

PLOS ONE

on behalf of Violet Naanyu (Academic Editor)

Journal Requirements:

Additional Editor Comments:

Consider the suggestion on use of visuals (Table) to communicate with the reader better.

Reviewers' comments:

Reviewer's Responses to Questions

**Comments to the Author**

1. If the authors have adequately addressed your comments raised in a previous round of review and you feel that this manuscript is now acceptable for publication, you may indicate that here to bypass the “Comments to the Author” section, enter your conflict of interest statement in the “Confidential to Editor” section, and submit your "Accept" recommendation.

Reviewer #3: All comments have been addressed

2. Is the manuscript technically sound, and do the data support the conclusions?

Reviewer #3: Yes

3. Has the statistical analysis been performed appropriately and rigorously? 

Reviewer #3: N/A

4. Have the authors made all data underlying the findings in their manuscript fully available?

Reviewer #3: Yes

5. Is the manuscript presented in an intelligible fashion and written in standard English?

Reviewer #3: Yes

6. Review Comments to the Author

Reviewer #3: All comments are now addressed to my satisfaction.

Would recommend including Table 2 provided in the response within the paper itself and referenced as such

Thank you

7. PLOS authors have the option to publish the peer review history of their article (what does this mean?). If published, this will include your full peer review and any attached files.

Reviewer #3: No

---

## [Author Response · Author response to Decision Letter 2]

24 Jul 2023

Reviewer #3: All comments are now addressed to my satisfaction. Would recommend including Table 2 provided in the response within the paper itself and referenced as such

Response: Thank you so much for the recommendation, table 2 has been included in the paper. See line number 291

---

## [Editor Report · Decision Letter 3]

13 Aug 2023

Finding Women in Fishing Communities around Lake Victoria: “feasibility and acceptability of using phones and tracking devices”.

PONE-D-22-04440R3

Dear Dr.Gertrude Nanyonjo,

We’re pleased to inform you that your manuscript has been judged scientifically suitable for publication and will be formally accepted for publication once it meets all outstanding technical requirements.

Kind regards,

Violet Naanyu, PhD

Academic Editor

PLOS ONE

---

## [Editor Report · Acceptance letter]

23 Aug 2023

PONE-D-22-04440R3 

Finding women in fishing communities around Lake Victoria: “feasibility and acceptability of using phones and tracking devices”. 

Dear Dr. Nanyonjo:

I'm pleased to inform you that your manuscript has been deemed suitable for publication in PLOS ONE. Congratulations! Your manuscript is now with our production department. 

Kind regards, 

on behalf of

Prof. Violet Naanyu 

Academic Editor

PLOS ONE